# Solar-Assisted Monetization of Municipal Solid Waste

Eman Aldamigh [1], Sarbajit Banerjee [2] and Mahmoud M. El-Halwagi [3,4,*]

[1] Department of Multidisciplinary Engineering, Texas A&M University, College Station, TX 77843, USA
[2] Department of Chemistry, Texas A&M University, College Station, TX 77843, USA
[3] Department of Chemical Engineering, Texas A&M University, College Station, TX 77843, USA
[4] Texas A&M Engineering Experiment Station Gas and Fuels Research Center, College Station, TX 77843, USA
[*] Correspondence: el-halwagi@tamu.edu

**Abstract:** Municipal solid waste (MSW) is a significant resource, especially for biomass-based monetization. In addition to its economic potential, it can also provide an effective pathway for decarbonization in the energy and chemical sectors. In this work, MSW monetization is coupled with the solar-assisted generation of "green" hydrogen and oxygen via electrolysis. The dual utilization of electrolysis-produced hydrogen and oxygen offers several advantages, including the reduction in the carbon footprint, the tunability of the characteristics of synthesis gas (syngas) to conform to the values needed for the manufacture of various chemicals, and the reduction in the overall cost compared to systems focusing on hydrogen generation only. A superstructure is developed to represent the configurations of interest and serve as a basis for formulating an optimization program that can be solved to identify the optimal design and operating strategies. A multi-period optimization formulation is developed to identify the maximum profit subject to the various modeling equations and constraints. The environmental considerations are addressed using the ε-constraint method by iteratively varying carbon footprint cuts. A case study is solved for the City of Jeddah, and the results assess the tradeoffs of various design and operating strategies, their impact on profitability, and their environmental impact.

**Keywords:** sustainability; biorefining; process integration; multi-period optimization; solar; green syngas

## 1. Introduction

Municipal solid waste (MSW) is generated in substantial quantities, with current estimates of more than 2 billion tons/year at present and the expectation to grow to more than 3 billion tons per year in 2050 [1,2]. Because of the utilizable components of MSW, there is a growing trend to use biorefining platforms, especially to valorize the organic ingredients [3,4]. This monetization approach involving processing steps to create value-added products is a major shift from the conventional approach of MSW disposal. Several strategies have been proposed for sustainable waste management, including physical, chemical, biological, and biochemical approaches such as incineration, pyrolysis, gasification, liquefaction, anaerobic digestion, and landfilling [5–7]. These approaches can lead to several benefits, including the following [8–11]:

- The substitution of fossil-based raw materials with renewable feedstocks for the production of energy, energy carriers, and chemicals. This substitution provides a reduction in the carbon footprint because of the lower emissions associated with bio-based feedstocks compared to fossil fuels.
- The mitigation of environmental and health hazards associated with the disposal of MSW in landfills. Such hazards include the release of landfill gases, which increase the carbon footprint and expose the public to undesirable emissions, the potential leaching into underground water, and the ineffective utilization of land.

- The integration of MSW into the circular economy leads to the conservation of natural resources, the development of cleaner technologies, and the creation of new jobs. These benefits can lead to a significant positive impact on sustainable development through the advancement of economic, environmental, and societal objectives.

In addition to biorefining, solar energy is an effective resource toward decarbonization. Solar-assisted systems have been proposed for various industrial applications. Relevant to this paper are the following solar-assisted areas:

- The production of synthesis gas (syngas) and chemicals [12]: Several feedstocks, such as MSW, natural gas, and coal, have a high level of chemical stability that makes them difficult to directly convert to value-added products. An important industrial step is reforming what converts these feedstocks to syngas (a mixture composed primarily of hydrogen, carbon monoxide, and carbon dioxide). Syngas is a readily reactive mixture that can be converted into a wide variety of chemicals and energy carriers. The utilization of solar energy to provide energy for the reforming process offers multiple benefits, including reducing the carbon footprint and enhancing sustainability. This approach also overcomes a key limitation in utilizing solar energy, which is the dynamic variability in collected energy due to the diurnal nature of solar systems. The use of solar energy in the production of syngas, which is subsequently converted into chemicals, serves as an elaborate method for storing variable solar energy in the form of chemical products and energy carriers that can be readily stored and dispatched as needed without being directly tied to the variability in collected solar energy. This is an important consideration in chemical manufacturing and energy transition because of the benefits of consistency and mobility.
- Cogeneration [13,14]: Solar energy may be collected in the forms of thermal energy and electric energy (e.g., through photovoltaic "PV" systems). Solar energy can also be integrated with combined heat and power (cogeneration) systems for the simultaneous production of thermal and electric energy. These systems can be operated using solar energy exclusively or may integrate solar energy with fossil fuels through dynamic schemes intended to provide a consistent output in spite of the variability in collected solar energy.
- Desalination and water–energy nexus [15,16]: Solar-assisted desalination is receiving growing attention, especially in arid areas with limited fresh water supplies. Fresh water is needed in several steps of the MSW monetization systems and the creation of green chemicals and energy carriers. For instance, water electrolysis is a major technology used in the production of green hydrogen. The use of solar energy in desalinating saline and brackish water offers several advantages, especially toward the reduction in the carbon footprint and the practicality in areas that are remote from the main power supply grids and infrastructure. Therefore, solar energy can be instrumental in enhancing synergistic applications of energy and water through a nexus approach that integrates both systems in a cost-effective and sustainable manner.

In this work, an integrated system is proposed for coupling solar energy with MSW monetization. A process integration framework is used to systematize the development and optimization of the system. Process integration [17] is a holistic framework for process synthesis, the conservation of mass and energy resources, the optimization of tasks and resources, design, and operations to create synergistic processing opportunities among various technologies and natural resources. In the proposed embodiment, green syngas is produced through a combination of solar-assisted water electrolysis, green oxygen gasification of MSW, syngas enrichment and turning with green hydrogen, and manufacturing of energy carriers and chemicals. A multi-period superstructure and optimization approach are developed to represent the integrated system and enable the development of optimal design and operational strategies. A case study on valorizing MSW in the City of Jeddah, Saudi Arabia, is solved to illustrate the merits of the proposed approach and the tradeoffs, especially in the economic and environmental objectives. The importance of this paper

stems from several factors. The superstructure representation embeds the plausible process configurations and enables the development of the optimization formulation, which aids in the methodical decision making. The introduction of a "green oxygen" pathway to aid in MSW gasification provides economic and environmental benefits. The case study and associated results provide valuable insights and potentials for applicability.

## 2. Problem Statement

Given is a specific region with an available supply of pre-sorted MSW (after removing metals, glass, electronic waste, building materials, etc.) that varies seasonally ($MSW_t^{Avail}$), where $t$ is an index representing a season/period. It is desired to monetize MSW into value-added chemicals using an integrated system of solar- and fossil-assisted energy, desalination, electrolysis, gasification, separation, and chemical manufacturing.

Available for service is an existing power plant integrated with a thermal desalination unit. In addition to the power generated by the power plant $\left(E_t^{PP\_Avail}\right)$, a PV solar system provides electric power $\left(E_t^{Solar\_Avail}\right)$. The desalinated water $\left(W_t^{Avail}\right)$ can be used in electrolysis to produce hydrogen and oxygen. The electric power and water used in electrolysis are bound by the availability constraints:

$$E_t^{PP} \leq E_t^{PP\_Avail} \ \forall t \tag{1}$$

$$E_t^{Solar} \leq E_t^{Solar\_Avail} \ \forall t \tag{2}$$

$$W_t^{Electrolysis} \leq W_t^{Avail} \ \forall t \tag{3}$$

The levelized prices of the power plant energy and solar energy are known ($LEC_t^{PP}$ and $LEC_t^{Solar}$, respectively) in units of (USD/MWh$_e$).

The MSW is gasified using the produced oxygen to generate syngas. The amount of MSW fed to the gasifier is bound by the availability constraints:

$$MSW_t \leq MSW_t^{Avail} \ \forall t \tag{4}$$

A chemical manufacturing process combines the syngas from the gasifier along with hydrogen from electrolysis to tune the characteristics of the syngas, including the H$_2$:CO ratio needed for chemical production and the relationships between the compositions of hydrogen, CO, and CO$_2$.

The emission factors from the electric energy used from the power plant and the solar system (*EF_PP$_t$* and *EF_Solar$_t$*, respectively) in units of (ton CO$_{2eq}$/MWh$_e$). Furthermore, the emission factors for the gasification and chemical manufacturing processes (*EF_Gas$_t$* and *EF_Chem$_t$*, respectively) in units of (ton CO$_{2eq}$/ ton syngas and ton CO$_{2eq}$/ton product, respectively).

The objective is to identify the optimum design and operation of the integrated system while responding to the following key questions:

- How much solar energy should be used in each period?
- How much water is needed per period?
- What is the optimal blending strategy of syngas produced from gasification and hydrogen generated from electrolysis?
- How do the optimal strategies vary under several carbon footprint constraints?

To address the aforementioned questions, an optimization-based approach will be developed. First, a structural representation will be introduced to combine the various resources and processing steps. Because of the dynamic/seasonal variabilities in solar energy and characteristics of MSW, a multi-period design and operation model will be used. The desired objectives, performance equations, limits and constraints, and available data will be coordinated through an optimization formulation and solved to generate the

necessary information for the design and operation of the system as well as the tradeoffs emanating from the conflicting natures of the economic and environmental objectives.

## 3. Optimization Formulation

The first step in the proposed approach is to create a representation that embeds the various embodiments of the process. Figure 1 illustrates a superstructure representation of the integrated system. It combines several building blocks for power collection and generation, water desalination and electrolysis, gasification of pre-sorted MSW, oxygen utilization in gasification, hydrogen blending for tuning the composition of syngas, and chemical manufacturing to convert the syngas into value-added products. Electric energy driving electrolysis is taken from a fossil-based power plant and a solar system. The desalination plant is thermally integrated with the power plant. Oxygen produced from electrolysis is used to gasify the MSW. This is a key advantage compared to most systems that focus on utilizing hydrogen produced from electrolysis without a high-value utilization of the produced oxygen. In the proposed system, both "green" hydrogen and oxygen are effectively utilized. The generated syngas is mixed with the hydrogen produced from electrolysis to provide the proper H$_2$:CO ratio needed for chemical manufacturing. This is another advantage of the system to enable the tuning of the syngas characteristics to any desired value depending on the intended chemical production. It is worth noting that this novel configuration is based upon the integration of mature building blocks. As such, the implementation of such systems should have a high level of technological maturity.

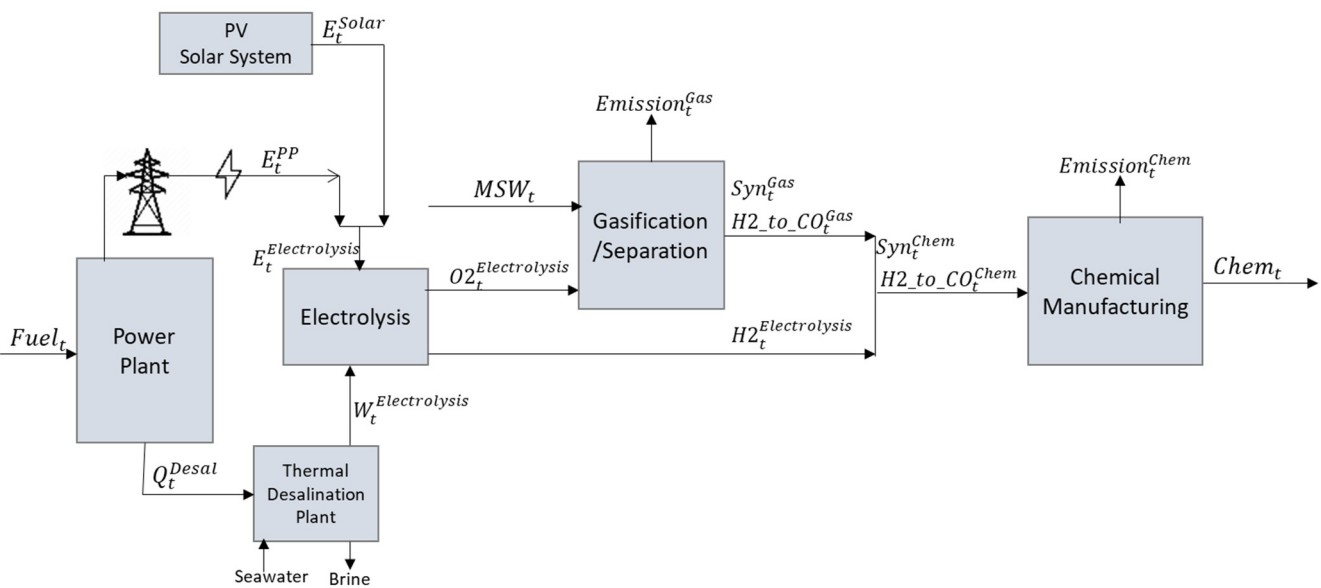

**Figure 1.** A superstructure representation of the integrated system.

The objective function may be a single economic objective or a multi-objective function encompassing several criteria (e.g., economic, environmental). For instance, a profit-based function is expressed as follows:

$$\text{Maximize Profit} = \sum_t Price_t^{Chem} \times Chem_t - TAC \tag{5}$$

where the sales revenue is the product of the amount of the main product(s) times the selling price of the product(s) and the total annualized cost is given by the summation of annualized fixed costs of electrolysis, gasification, and chemical manufacturing, the annual operating costs of gasification and chemical manufacturing, and the levelized costs of solar and power-plant energies, and the costs of water and pre-sorted MSW, i.e.:

$$TAC = AFC^{Electrolysis} + AFC^{Gas} + AFC^{Chem} + \sum_t (LEC_t^{Solar} \times E_t^{Solar} + LEC_t^{PP} \times E_t^{PP} + Price_t^{Water} \times W_t^{Electrolysis} + Price_t^{MSW} \times MSW_t + AOC_t^{Gas} + AOC_t^{Chem})$$

The following main constraints are used:

Energy input to electrolysis come from the solar system and/or power plant:

$$E_t^{Electrolysis} = E_t^{Solar} + E_t^{PP} \; \forall t \tag{6}$$

This energy input to electrolysis is related to the amount of water to be electrolyzed via a factor $\alpha^{Electrolysis}$. Therefore,

$$E_t^{Electrolysis} = \alpha^{Electrolysis} \times W_t^{Electrolysis} \; \forall t \tag{7}$$

Based on stoichiometric dissociation of water, the mass balances for the produced hydrogen and oxygen are given by the following:

$$H2_t^{Electrolysis} = \frac{1}{9} \times W_t^{Electrolysis} \; \forall t \tag{8}$$

$$O2_t^{Electrolysis} = \frac{8}{9} \times W_t^{Electrolysis} \; \forall t \tag{9}$$

Gasification and chemical manufacturing process models:

$$\left( Syn_t^{Gas}, \; H2\_to\_CO_t^{Gas}, Emission_t^{Gas} \right) = \Omega \left( MSW_t, O2_t^{Electrolysis} \right) \; \forall t \tag{10}$$

where $\Omega$ is a set of the gasifier's process equations relating the syngas yield, $H_2$:CO ratio, and emissions to the flowrate and characteristics of the MSW, the ratio of oxygen to MSW, and technology type, design, and operation. Similarly, the chemical manufacturing process can be modeled using a set of equations, $\Psi$, relating the product flowrate and emissions to the feed, design, and operation:

$$\left( Chem_t, Emission_t^{Chem} \right) = \Psi \left( Syn_t^{Chem}, H2\_to\_CO_t^{Chem} \right) \forall t \tag{11}$$

Availability constraints include maximum available energy from the power plant, the solar system, and the maximum available amounts of the desalinated water and the pre-sorted MSW were given by Equations (1)–(4).

Market limitations:

$$Chem_t \leq Chem_t^{max} \; \forall i \tag{12}$$

$H_2$:CO ratio needed for chemical manufacturing:

$$H2\_to\_CO_t^{Chem} \geq H2\_to\_CO_t^{min} \; \forall t \tag{13}$$

The ε-constraint method is used to account for the environmental objective:

$$\sum_t Emission_t \leq Emission^{Max} \tag{14}$$

where the annual emissions are calculated using the following equation:

$$Emission_t = \sum_t (EF\_PP_t * E_t^{PP} + EF_{Solar t} * E_t^{Solar} + Emission_t^{Gas} + Emission_t^{Chem}) \tag{15}$$

Equations (1)–(15) constitute the generic formulation of the optimization formulation. It is a nonlinear program that can be solved to identify the design and operation of the

building blocks in the process and to determine the extent of solar energy used in each period, the flows of mass and energy throughout the system, the emissions of the system, and the optimal values of the objectives. Because of the need to reconcile the economic and environmental objectives (e.g., the annual profit or return on investment versus carbon footprint), the ε-constraint method is used by iteratively varying carbon footprint cuts and identifying the optimal economic objective at each cut. The result is a tradeoff (Pareto) curve that can aid the decision-makers in selecting appropriate designs while considering the economic and environmental objectives.

## 4. Case Study

The case study is aimed at monetizing a portion of MSW generated in the city of Jeddah, Saudi Arabia, into value-added methanol. The city of Jeddah has a population of about 4.7 MM. An influx of visitors increases the population, especially during the four-month period around the season of Ramadan, Eid-ul-Fitr, and Eid-ul-Adha. Therefore, two periods are considered in this case study: (t = 1) is an eight-month period with an MSW production [18] of about 1.24 kg/person/day (or 5800 tons/day), and (t = 2) is a four-month period with an estimated increase of about 20% in the daily MSW production. The recyclable content of the MSW that can be extracted to make refuse-derived fuel (RDF) [19] with practical recovery ranges from 64% to 77% [20]. It is anticipated that 15–20% of the MSW recyclable materials can be practically recovered [21] and delivered in the form of RDF, which can then be fed to the gasifier. Therefore, the maximum flowrates of RDF available as feed to the gasifier are 711 and 853 tons/day for periods 1 and 2, respectively. The primary product of the system is methanol. Compared to RDF incineration to produce thermal energy/electricity, conversion to methanol has the advantages of producing an easily transportable fuel/chemical and the monetization to a higher value-added product. The production of methanol from RDF also enjoys the advantage of a ~40% reduction in the carbon footprint compared to the production of methanol from fossil fuels [22].

The key modeling equations and simulation data for methanol synthesis are taken from several sources [22–26]. Furthermore, the following data are used to formulate the optimization model:

- Gasifier feed: $O_2$:RDF mass ratio = 0.42 [27].
- Syngas products: Syngas: RDF mass ratio = 0.63. Composition of syngas (mol%): 42% $H_2$, 42% CO, 8% $CO_2$, and 8% $H_2O$ [27,28].
- Methanol production: Feed requirements: The stoichiometric number S [23]:

$$S = \frac{Moles\ H_2 - Moles\ CO_2}{Moles\ CO + Moles\ CO_2} \geq 2.0 \qquad (16)$$

- Molar yield of methanol = 85% of stoichiometric target [23,26].
- Electrolysis: Electric energy needed for electrolysis: 16,459 MJ/ton of water by assuming 80% efficiency of the theoretical target of 13,167 MJ/ton of water [29].
- Carbon footprint:
  - Combined gasification and methanol manufacture: 1.7 tons of $CO_{2eq}$: ton of methanol [25,28].
  - Electric energy used in electrolysis: Emission factors: 0.54 and 0.06 tons of $CO_{2eq}$/MWh$_e$ for electric energy from the power plant using natural gas and from solar systems, respectively [16,30].
- Economic data:
  - Fixed capital investment (FCI) of electrolysis = USD 0.5 MM/MW$_e$ [31]

FCI of gasifier and methanol manufacture = FCI (in USD MM) = 0.16 × N × (Flowrate of biomass feed in 1000 tons/yr)$^2$  (17)

where N is number of functional steps (=6 in this case study) [25].

o  Annualized fixed cost (AFC) is calculated using a 10-year linear depreciation scheme with a salvage value equal to 30% of the initial value.

o  Working capital investment (WCI) = 20% of FCI [32].

o  Total capital investment:

$$TCI = FCI + WCI \tag{18}$$

o  RDF: USD 50/ton.

o  Non-feedstock operating cost for methanol manufacture = USD 22.6 ton/methanol [26].

o  Cost of desalinated water: USD 3/ton.

o  Levelized cost of electric energy: USD 40/MWhe from a power plant using natural gas and USD 75/MWhe from the solar plant.

o  Selling price of methanol: USD 500/ton.

The optimization program was formulated with the objective of maximizing the annual profit subject to the aforementioned constraints and data. Furthermore, the return on investment (ROI) was also calculated according to the following expression [32]:

$$ROI = \frac{Annual\ Profit}{TCI} \times 100\% \tag{19}$$

The ε-constraint method was used to account for the solution under various carbon footprint limits. The resulting nonlinear program with 62 constraints and 71 variables was repeatedly solved for specific carbon footprint cuts using the Global Solver of the software LINGO. First, the optimization program was solved without a carbon footprint limit. A maximum profit of USD 26.994 MM/year was obtained with a total carbon footprint of 552,858 tons of $CO_{2eq}$/year. Next, the carbon footprint was iteratively reduced to generate the tradeoff curves and associated solutions. Figure 2 is an example of the generated solutions. It represents the maximum profit solution when the carbon footprint limit is set to 400,000 tons of $CO_{2eq}$/year. Figure 3 shows the impact of changing the carbon footprint limit on the percentage contribution of the solar energy to the total energy consumed in electrolysis. When all the energy is obtained from the power plant, the total emissions are 552,858 tons of $CO_{2eq}$/year. With the gradual introduction of solar energy, the carbon footprint progressively decreases, as shown in Figure 3. On the other hand, as the carbon footprint limit is tightened, the annual profit and return on investment decrease, as shown in Figures 4 and 5, respectively. For instance, when there are no limits on the carbon footprint, the annual profit is USD 27 and the ROI is 15%/year. When the carbon footprint is limited to a maximum of 350,000 tons of $CO_{2eq}$/year, the annual profit is USD 12 and the ROI is 7%/year. Depending on the desired ROI (which varies from one investor to another), a decision can be made on the extent of the solar energy contribution and carbon footprint. For instance, if the minimum acceptable ROI is 11%/year, the solar energy contributes 36% of the total energy input and the carbon footprint is 450,000 tons of $CO_{2eq}$/year.

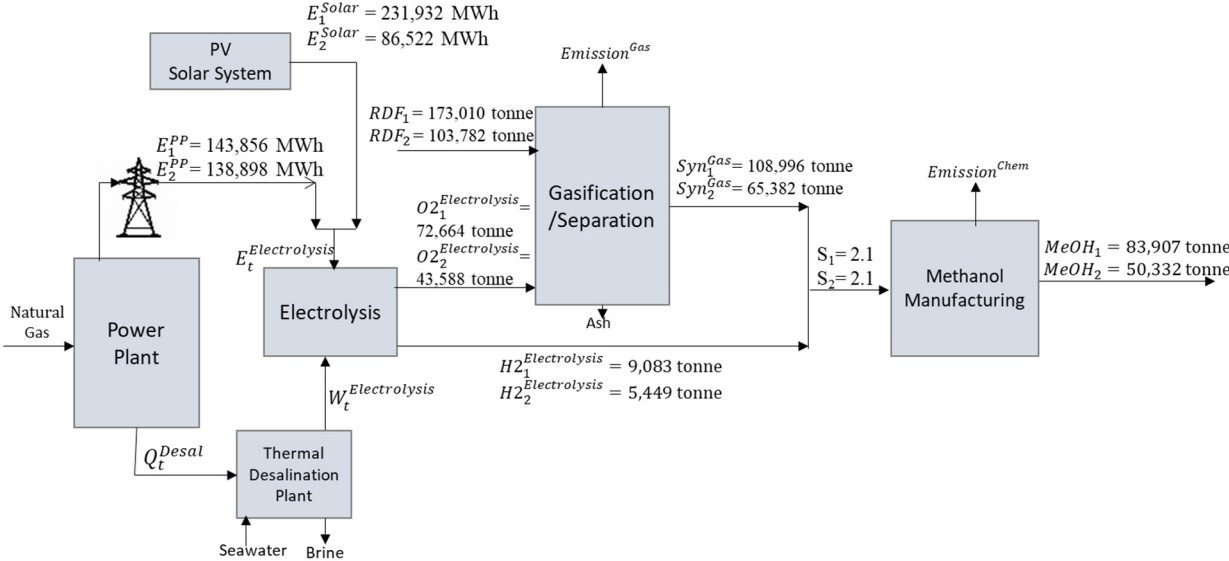

**Figure 2.** Optimal solution under a carbon footprint limit of 400,000 tons of $CO_{2eq}$/year.

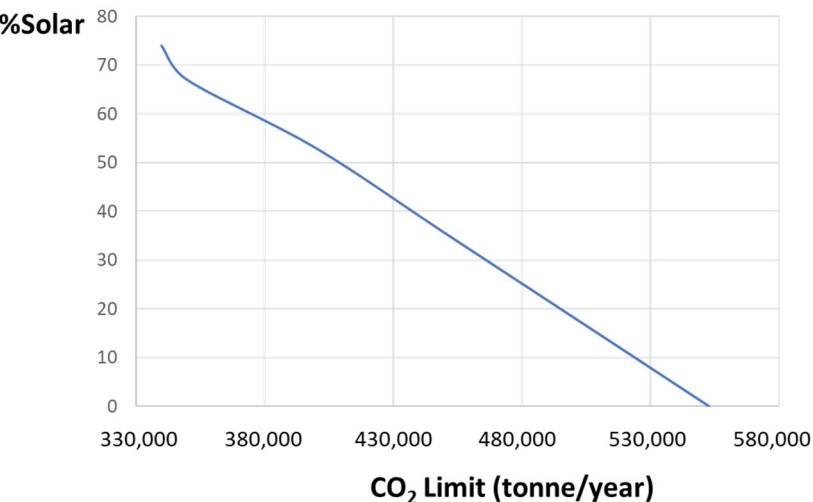

**Figure 3.** Impact of the carbon footprint limit on the percentage contribution of solar energy to the electric energy use in electrolysis.

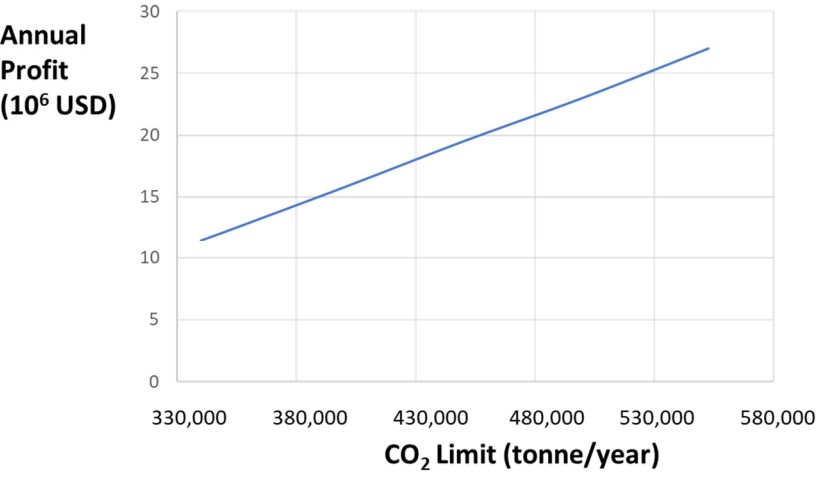

**Figure 4.** Impact of the carbon footprint limit on maximum annual profit.

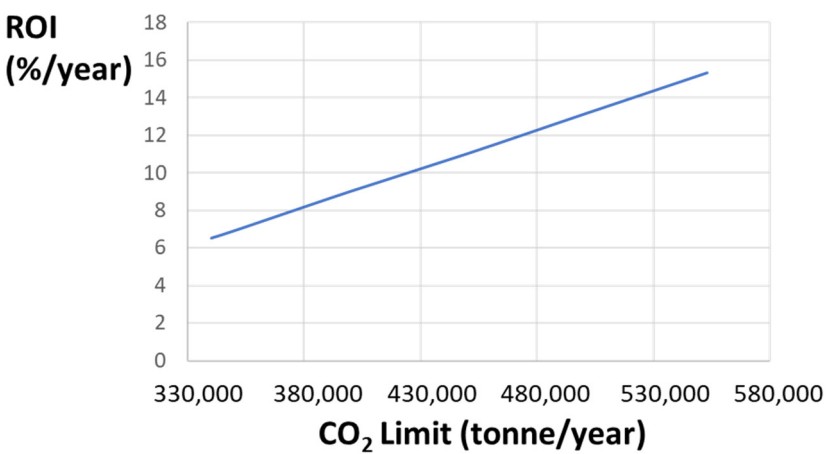

**Figure 5.** Impact of the carbon footprint limit on return on investment.

### 5. Conclusions and Recommendations for Future Work

This work has introduced an integrated approach to incorporating solar energy and green hydrogen and oxygen in monetizing MSW into value-added products. The pre-sorted MSW is used after the separation of recoverable and non-processible ingredients such as metal, glass, electronic waste, building materials, etc. Gasification of the pre-sorted MSW is carried out using oxygen to produce syngas that can be converted to value-added chemicals and energy carriers. Solar energy is integrated with fossil energy for electrolysis and desalination. The desalinated water is electrolyzed to hydrogen and oxygen. The oxygen is used in gasifying MSW, while the hydrogen is used for tuning the quality of the produced syngas. This novel configuration offers several advantages, including the following:

- The integration of solar energy and fossil fuels ensures reliable operation in spite of the dynamic variability in solar intensity.
- The production of hydrogen and oxygen through solar-assisted electrolysis reduces the carbon footprint and serves to indirectly store solar energy in the form of chemicals that can be readily stored and dispatched.
- The dual usage of hydrogen and oxygen produced from electrolysis enhances the value proposition of the proposed system.
- The production of syngas from MSW and green hydrogen and oxygen warrants the designation of "green syngas", which can then be used to produce "green" chemicals and energy carriers.
- Blending the syngas produced from gasification with hydrogen produced from electrolysis offers high levels of tunability of syngas characteristics for optimal utilization in the manufacture of a wide variety of chemicals.
- The modular nature of most building blocks in the process enhances the ease of the expanding capacity by simply adding more modules or shrinking the capacity by deploying some of the modules to other regions on a need basis.
- The high levels of mass and energy integration within the system lead to synergistic opportunities, cost reduction, resource conservation, pollution prevention, and enhanced sustainability.

A superstructure-based representation and the associated optimization formulation were developed. The model equations included the performance functions for the key building blocks of the process, energy and mass balances, allocation, and tracking, as well as availability constraints (e.g., energy, supply of MSW, market demands for products). An economic objective function (annual profit or return on investment) and an environmental objective function (carbon footprint) were used and reconciled via the ε-constraint method by iteratively changing the carbon footprint limits and determining the optimal economic objective at each environmental limit. The result is a tradeoff (Pareto) curve that can

aid decision-makers in selecting appropriate designs while considering economic and environmental objectives.

A case study was solved to monetize MSW generated in Jeddah, Saudi Arabia, into value-added methanol. The optimization program was solved under different carbon footprint limits. The generated solutions showed the tradeoffs of carbon footprint versus the optimal characteristics including percentage contribution of solar energy, profit, and return on investment.

Recommendations for future research include the following:

- The inclusion of safety as a key objective in addition to the economic and environmental objectives. Inherently safer design techniques and metrics [25,33–35] can be included in the developed optimization formulation. It is also worth noting that the economic, environmental, and safety objectives can be combined through a profitability framework using the concept of safety- and sustainability-weighted return on investment [36,37].
- The consideration of resilience of the proposed system in the context of an integrated supply chain and the potential for failures resulting from natural disasters and abnormal disruptions [38–40]. The key objective is for the system to mitigate the risks of failure, and if it fails, it can recover rapidly and with minimal impact on associated communities.
- The incorporation of natural gas in the system. Natural gas is a principal source for the production of syngas [41,42]. As such, significant economic, environmental, and reliability benefits can accrue as a result of integrating natural gas reforming with MSW gasification. Furthermore, such integration will also pave the way to the co-production of syngas and solid carbon (e.g., highly valuable multi-walled carbon nanotubes) [43–45].
- The use of landfill gases, biogas, and flare gas [46] in the system to enhance syngas production and reduce the carbon footprint.
- Integration into a distributed manufacturing network. Recent approaches in biorefining enable the synergistic coupling of systems like the one introduced in this paper with a network of biomass suppliers, decentralized infrastructures, and product distribution through a distributed manufacturing platform [25,47].
- The development of skid-mounted modular implementations of the proposed system that can be deployed to different regions on a need basis. Applications of this concept to analogous systems monetizing gas have shown several economic and environmental advantages [48].
- Timed use of excess solar energy. Because of the variable nature of solar energy, the collected energy may exceed the capacity of the grid. A common approach to resolving this issue is "curtailment", which corresponds to the reduction in the amount of collected solar energy to ensure grid integrity and maintain a proper balance between energy supply and demand [49]. This offers an opportunity to use the excess solar energy (available at a reduced price to avoid curtailment) in MSW monetization.
- An expansion of the case study to develop strategic plans. The framework and optimization tools introduced in this work can be applied to set national strategies for regions and countries and provide guidelines for decarbonization initiatives.

**Author Contributions:** Conceptualization, E.A., S.B. and M.M.E.-H.; Methodology, E.A., S.B. and M.M.E.-H.; software, E.A. and M.M.E.-H.; validation, E.A. and M.M.E.-H.; data curation, E.A., S.B. and M.M.E.-H.; writing—E.A. and M.M.E.-H.; writing—S.B. and M.M.E.-H.; supervision, S.B. and M.M.E.-H.; project administration, M.M.E.-H. All authors have read and agreed to the published version of the manuscript.

**Funding:** Aldamigh received funding from Kingdom of Saudi Arabia.

**Conflicts of Interest:** The authors declare no conflict of interest.

## Nomenclature

| | |
|---|---|
| *AFC* | Annualized fixed cost |
| *AOC* | Annual operating cost |
| $\textbf{\textit{Chem}}_t$ | Amount of produced chemical over period t |
| $E_t^{Electrolysis}$ | Electric energy used in electrolysis over period t |
| $E_t^{PP}$ | Electric energy from the power plant used in electrolysis over period t |
| $E_t^{PP\_Avail}$ | Available energy from the power plant over period t |
| $E_t^{Solar}$ | Electric energy from the solar system used in electrolysis over period t |
| $E_t^{Solar\_Avail}$ | Available energy from the solar system over period t |
| $EF\_Chem_t$ | Emission factor for chemical manufacturing during period t |
| $EF\_Gas_t$ | Emission factor for the gasification unit during period t |
| $EF\_PP_t$ | Emission factor for the electric energy from the power plant during period t |
| $EF\_Solar_t$ | Emission factor for the electric energy from the solar system during period t |
| $\textbf{\textit{Emission}}_t$ | Emission amount over period t |
| *FCI* | Fixed capital investment |
| *LEC* | Levelized energy cost |
| $\textbf{\textit{MSW}}_t$ | The amount of the MSW fed to the gasifier over period t |
| $\textbf{\textit{MSW}}_t^{Avail}$ | Available amount of MSW over period t |
| $\textbf{\textit{Price}}_t^{Chem}$ | Selling price of the produced chemical during period t |
| $\textbf{\textit{Price}}_t^{Water}$ | Price of water during period t |
| $Syngas_t$ | Amount of syngas produced over period t |
| *TAC* | Total annualized cost |
| *WCI* | Working capital investment |
| $W_t^{Avail}$ | Available water over period t |

**Subscripts**

| | |
|---|---|
| *t* | Time period |

**Superscripts**

| | |
|---|---|
| *Avail* | Available |
| *Chem* | Chemical manufacture |
| *Electrolysis* | Electrolysis system |
| *Gas* | Gasification |
| *Max* | Maximum |
| *Min* | Minimum |
| *PP* | Power plant |
| *Solar* | Solar energy |

**Greek**

| | |
|---|---|
| $\alpha^{Electrolysis}$ | Factor relating energy usage to water electrolysis |
| $\Psi$ | Set of equations relating the product flowrate and emissions to the feed, design, and operation |
| $\Omega$ | Set of the gasifier's process equations relating the syngas yield, H2:CO ratio, and emissions to the flowrate and characteristics of the MSW, the ratio of oxygen to MSW, and technology type, design, and operation |

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
