# Peer review of "Solar-Assisted Monetization of Municipal Solid Waste"

_processes, doi:10.3390/pr11072174_

Round 1

Reviewer 1 Report

Municipal solid waste (MSW), as a daily human by-product, is critical to both social development and the environment. Municipal solid waste (MSW) incineration can generate significant amounts of energy to replace coal-fired power generation. The paper explores the possibility of monetizing MSW with solar-assisted electrolysis to produce "green" hydrogen and oxygen, and calculates the energy consumption and carbon emissions, which is very interesting work.
1. the importance of the paper needs to be further enhanced in the introduction. 2.
2. The article proposes that "Solar-Assisted Monetization of Municipal Solid Waste", the feasibility of the technology should focus on the discussion.

3. The MSW contains ingredients such as kitchen waste, plastic, rubber and paper, and whether the MSW needs further treatment to be treated.
4. Line 113, H2 should be H2.
5. Compared with MSW incineration, the prospect and applicability of this technology.

The English language is fine.

Author Response

Response to Reviewer #1’s Comments:

Comment: Municipal solid waste (MSW), as a daily human by-product, is critical to both social development and the environment. Municipal solid waste (MSW) incineration can generate significant amounts of energy to replace coal-fired power generation. The paper explores the possibility of monetizing MSW with solar-assisted electrolysis to produce "green" hydrogen and oxygen, and calculates the energy consumption and carbon emissions, which is very interesting work.

Response: We thank the reviewer for the valuable comments and feedback.

Comment 1. the importance of the paper needs to be further enhanced in the introduction. 2.

Response: We have expanded the introduction to describe the key important contributions of the paper. Specifically, we discussed: The superstructure representation embeds the plausible process configurations and enables the development of the optimization formulation which aids in the methodical decision making. The introduction of a “green oxygen” pathway to aid in MSW gasification provides economic and environmental benefits. The case study and associated results provide valuable insights and potential for applicability. 

Comment 2. The article proposes that "Solar-Assisted Monetization of Municipal Solid Waste", the feasibility of the technology should focus on the discussion.

Response: We have emphasized the feasibility and especially that this novel configuration is based upon the integration of mature building blocks. As such, the implementation of the such systems should have a high level of technology maturity.

Comments: 3. The MSW contains ingredients such as kitchen waste, plastic, rubber and paper, and whether the MSW needs further treatment to be treated.

Response: This is indeed an important point. We have clarified that the feed to the gasifier is pre-sorted MSW after recovering metals, glass, electronic waste, building materials, etc.

Comment 4. Line 113, H2 should be H2.

Response: Since H2 is part of the variable name, we kept it as H2.

Comment 5. Compared with MSW incineration, the prospect and applicability of this technology.

Response: We have added the comparison. Specifically, we mentioned that compared to RDF incineration to produce thermal energy/electricity, conversion to methanol has the advantages of producing an easily transportable fuel/chemical and the monetization to a higher val-ue-added product. The production of methanol from RDF also enjoys the advantage of ~40% reduction in the carbon footprint compared to the production of methanol from fossil fuels.

Reviewer 2 Report

The reviewed scientific paper titled "Solar-Assisted Monetization of Municipal Solid Waste" presents an interesting and potentially valuable contribution to the field. The paper explores the dual utilization of electrolysis-produced hydrogen and oxygen from MSW, highlighting its economic potential and its role in decarbonization efforts in the energy and chemical sectors.

However, there are several areas of concern that need to be addressed before the paper can be considered for publication. The main issues are as follows:

1. The methodology employed in the study lacks clarity and detail. It is essential to provide a comprehensive description of the experimental setup, data collection, and analysis procedures. 

2. The paper fails to provide precise and comprehensive explanations of the formulas used. It is crucial to describe the mathematical models and equations in detail, including the underlying assumptions and parameters. 

3. Figures 3-5 are mentioned in the text, but no descriptions or explanations of their importance and contribution to the article are provided. Figures play a vital role in conveying information effectively, so it is essential to include clear and concise captions that explain the content and significance of each figure. 

The scientific paper presents a promising approach for MSW monetization and solar-assisted green hydrogen and oxygen generation. However, the paper's methodology requires clarity, the formulas need comprehensive explanations, and the figures require proper descriptions. These changes, along with others needed for publication, will strengthen the paper and enhance its contribution to the field.

 Minor editing of English language required

Author Response

Response to Comments from Reviewer #2:

Comment: The reviewed scientific paper titled "Solar-Assisted Monetization of Municipal Solid Waste" presents an interesting and potentially valuable contribution to the field. The paper explores the dual utilization of electrolysis-produced hydrogen and oxygen from MSW, highlighting its economic potential and its role in decarbonization efforts in the energy and chemical sectors.

Response: We appreciate the valuable comments provided by the reviewer.

Comment: However, there are several areas of concern that need to be addressed before the paper can be considered for publication. The main issues are as follows:

Comment 1. The methodology employed in the study lacks clarity and detail. It is essential to provide a comprehensive description of the experimental setup, data collection, and analysis procedures. 

Response: No experimental work was carried out in this study, Nonetheless, we have expanded the description of the superstructure approach, the optimization formulation, and solution strategy. We have also described the sources of the collected data.

Comments 2. The paper fails to provide precise and comprehensive explanations of the formulas used. It is crucial to describe the mathematical models and equations in detail, including the underlying assumptions and parameters. 

Response: We have expanded the description of the formulas and equation models with justification for background.

Comment 3. Figures 3-5 are mentioned in the text, but no descriptions or explanations of their importance and contribution to the article are provided. Figures play a vital role in conveying information effectively, so it is essential to include clear and concise captions that explain the content and significance of each figure. 

Response: We have expanded the discussions and insights for the figures and the significance of the results.

Round 2

Reviewer 2 Report

The paper has made improvements based on the provided comments. The authors have addressed several key areas, including the expansion of the description of the superstructure approach, the optimization formulation, and the solution strategy. This expansion will likely enhance the readers' understanding of the proposed methodology and its implementation.

Moreover, the authors have taken the feedback into account and expanded the description of the formulas and equation models presented in the paper. By providing more detailed explanations and context, they have made the mathematical aspects of their work more accessible to the readers. This clarity will contribute to the reproducibility and applicability of the research.

The authors have made efforts to expand the discussions and insights related to the figures presented in the paper. By elaborating on the significance of the results and providing additional insights, they have enhanced the overall interpretation and impact of their findings. This expansion of discussions will help the readers grasp the implications of the results and potentially inspire further research in the field.

Minor editing of English language required

Author Response

(The authors gave the same response as above.)
